# MULAN: A Blind and Off-Grid Method for Multichannel Echo Retrieval

**Helena Peić Tukuljac**
Department of Computer and Communication Sciences
École polytechnique fédérale de Lausanne
`helena.peictukuljac@epfl.ch`

**Antoine Deleforge**
Université de Lorraine, CNRS, Inria, LORIA
F-54000 Nancy, France
`antoine.deleforge@inria.fr`

**Rémi Gribonval**
Univ Rennes, Inria, CNRS, IRISA
35000 Rennes, France
`remi.gribonval@inria.fr`

## Abstract

This paper addresses the general problem of *blind echo retrieval*, *i.e.*, given $M$ sensors measuring in the discrete-time domain $M$ mixtures of $K$ delayed and attenuated copies of an unknown source signal, can the echo locations and weights be recovered? This problem has broad applications in fields such as sonars, seismology, ultrasounds or room acoustics. It belongs to the broader class of blind channel identification problems, which have been intensively studied in signal processing. Existing methods in the literature proceed in two steps: (i) blind estimation of sparse discrete-time filters and (ii) echo information retrieval by peak-picking on filters. The precision of these methods is fundamentally limited by the rate at which the signals are sampled: estimated echo locations are necessary *on-grid*, and since true locations never match the sampling grid, the weight estimation precision is impacted. This is the so-called *basis-mismatch* problem in compressed sensing. We propose a radically different approach to the problem, building on the framework of finite-rate-of-innovation sampling. The approach operates directly in the parameter-space of echo locations and weights, and enables near-exact blind and *off-grid* echo retrieval from discrete-time measurements. It is shown to outperform conventional methods by several orders of magnitude in precision.

## 1 Introduction

When a wave propagates from a point source through a medium and is reflected on surfaces before reaching sensors, the measured signals consist of mixtures of the direct path signal with delayed and attenuated copies of itself. This physical phenomenon is commonly referred to as *echoes* and has a wide range of applications in different areas of science, from sonars [1] to seismology [2], from acoustics [3, 4, 5] to ultrasounds [6]. For instance, in acoustics, it has been shown that precise knowledge of early echo timing enables the estimation of the positions of reflective surfaces in a room [3, 4]. In [3], the approximate 3D geometry of Lausanne cathedral could be retrieved in this way. On the other hand, echoes' attenuation capture information about the acoustic *impedance* of surfaces, which is notoriously hard to measure or estimate in practice [7, 8]. In [9] and [10], it is shown that knowing the attenuation and timing of early echoes may improve beamforming and source separation performance, respectively. Systems using echoes for beamforming are commonly referred to as *rake receivers* in the wireless literature [11].

Retrieving echo properties when the emitted signal is known is referred to as *active echolocation* in biology, and is well exemplified by the sensory system of echoing bats. This principle is for instance

at the heart of active sonar technologies. In the signal processing literature, this problem belongs to the category of *system* or *channel identification*, *i.e.*, estimating the filters from a known input to the observed output of a linear system. In the case of echoes, these linear filters consist of streams of Diracs in the continuous-time domain and are hence sparse in the discrete-time domain. The more challenging problem of estimating echoes/filters when the emitted signal is unknown is referred to as *passive echolocation* or *blind system identification* (BSI) [12, 13, 14, 15, 16, 17, 18, 5, 19, 20]. BSI is a long-standing and still active research topic in signal processing, notably due to its fundamental ill-posedness. In the general setting of arbitrary signals and filters, rigorous theoretical ambiguities under which the problem is unsolvable have been identified [12]. A number of methods for multichannel BSI with general signals and filters have been developed some time ago [12, 13, 14]. Some well-known limitations of these approaches are their sensitivity to the chosen length of filters, and their intractability when the filters are too large. Following the compressed sensing wave [21], a number of methods extending these BSI methods to the case of sparse [15, 16, 17, 18, 5] or structured [20] filters have been developed. They generally extend classical methods using regularizers such as the $\ell_1$-norm for sparsity or a bilinear constraint as in [20]. Similarly to classical filter estimation methods, they require knowledge of the filters' length and they work in the space of discrete-time filters which are typically thousands of samples long. Because they work in the discrete-time domain, the accuracy at which these methods can recover echo locations is fundamentally limited by the signal's frequency of sampling: the recovered echoes are *on-grid*. Moreover, the sparsity assumption on filters is invalid in practice due to smoothing and sampling effects at sensors. Interestingly, [22] employs a continuous-time spike model for single-channel blind deconvolution but relies on a strong linear prior on the signal.

In this paper, we propose a drastically different approach to blind echo retrieval based on the framework of finite-rate-of-innovation (FRI) sampling [23, 24, 25]. In stark contrast with existing methods, the approach directly operates in the space of continuous-time echoes, and is hence able to blindly recover their locations *off-grid*. The proposed method is shown to recover echo delays and attenuation with an accuracy far higher than what the sampling rate would normally allow, using noiseless multichannel discrete-time measurements of an unknown simulated speech emitter in a room. The method does not assume that the filters are finite-length and only requires the number of echoes. The remainder of this paper is organized as follows. In section 2 the signal and measurement models and notations are introduced, and conventional methods are briefly reviewed. In section 3 the proposed approach is presented in the non-blind and blind cases. In section 4 the method is compared and evaluated on both synthetic and room acoustic data. Conclusions and future directions are outlined in section 5.

## 2  Background

### 2.1  The signal and measurement models

We start by defining the signal model in the continuous-time domain. Suppose a source emits a band-limited signal $\widetilde{s}(t)$ which is reflected and attenuated $K$ times before reaching $M$ sensors. The continuous signal impinging at sensor $m$ is

$$\widetilde{x}_m(t) = (\widetilde{h}_m * \widetilde{s})(t) \tag{1}$$

where $\widetilde{h}_m$ is a linear filter from the source to sensor $m$ and $*$ denotes the continuous convolution operator defined by

$$(x * y)(t) = \int_{-\infty}^{+\infty} x(u)y(t-u)du. \tag{2}$$

The filter consists of the following stream of Diracs:

$$\widetilde{h}_m(t) = \sum_{k=1}^{K} c_{m,k}\delta(t - \tau_{m,k}), \tag{3}$$

where $\delta$ denotes the Dirac delta function, $\{\tau_{m,k}\}_{k=1}^{K}$ denote the $K$ propagation times from the source to sensor $m$ in seconds, *i.e.* the *echo delays* or Dirac locations and $\{c_{m,k}\}_{k=1}^{K}$ denote the *echo attenuations* or Dirac weights. In practical applications, continuous time-domain signals are not accessible. They are measured by sensors and discretized to be stored in a computer's memory.

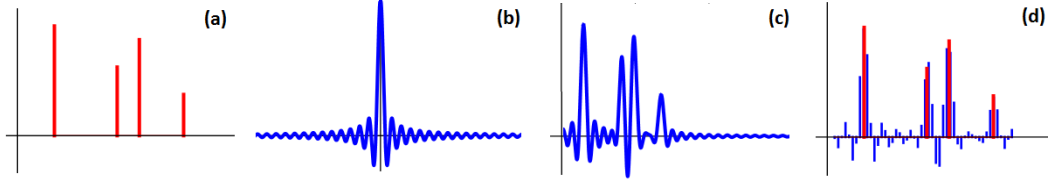

Figure 1: (a) Continuous-time stream of Diracs $\widetilde{h}(t)$, (b) sinc kernel $\widetilde{\phi}(t)$, (c) smoothed stream $(\widetilde{\phi} * \widetilde{h})(t)$, (d) original stream $\widetilde{h}(t)$ (red) and its smoothed, sampled version $\hat{h} \in \mathbb{R}^L$ (blue).

Let $\hat{\boldsymbol{x}}_m \in \mathbb{R}^N$ denote $N$ consecutive discrete samples collected by sensor $m$. Most measurement models assume that the impinging signal undergoes an ideal low-pass filter with frequency support $[-F_s/2, F_s/2]$ before being regularly sampled at the rate $F_s$ in Hz. This is expressed by

$$\hat{x}_m(n) = (\widetilde{\phi} * \widetilde{x}_m)(n/F_s), \ n = 0, \dots, N-1 \tag{4}$$

where $\widetilde{\phi} = \sin(\pi t)/\pi t$ is the classical sinc sampling Kernel. The continuous-time model (1) can then be approximated in two different ways, described in the next two sub-sections.

### 2.1.1 Discrete time-domain model

First, model (1) can be approximated in the discrete, finite-time domain. Let $\hat{\boldsymbol{h}}_m \in \mathbb{R}^L$ and $\hat{\boldsymbol{s}} \in \mathbb{R}^{N+L-1}$ denote discrete, sampled versions of the filter $\widetilde{h}_m$ and signal $\widetilde{s}$ respectively. We then have

$$\hat{x}_m(n) \approx (\hat{\boldsymbol{h}}_m \star \hat{\boldsymbol{s}})(n) \tag{5}$$

where the discrete finite convolution operator $\star$ between two vectors $\boldsymbol{u} \in \mathbb{R}^L$ and $\boldsymbol{v} \in \mathbb{R}^D$ ($L \leq D$) is defined by

$$(\boldsymbol{u} \star \boldsymbol{v})(n) = \sum_{j=0}^{L-1} u(j)v(L-1+n-j), \ n = 0, \dots, D-L. \tag{6}$$

The following convenient matrix notation will be used in the paper:

$$\boldsymbol{u} \star \boldsymbol{v} = \textit{Toep}_0(\boldsymbol{u})\boldsymbol{v} = \textit{Toep}(\boldsymbol{v})\boldsymbol{u} = \tag{7}$$

$$\begin{bmatrix} u_L & \dots & u_1 & 0 & \dots & \dots & 0 \\ 0 & u_L & \dots & u_1 & 0 & \dots & 0 \\ \vdots & \ddots & \ddots & \ddots & \ddots & \ddots & \vdots \\ 0 & \dots & 0 & \ddots & \ddots & \ddots & 0 \\ 0 & \dots & \dots & 0 & u_L & \dots & u_1 \end{bmatrix} \begin{bmatrix} v_1 \\ v_2 \\ \vdots \\ v_D \end{bmatrix} = \begin{bmatrix} v_L & v_{L-1} & \dots & v_1 \\ v_{L+1} & v_L & \ddots & v_2 \\ \vdots & \ddots & \ddots & \vdots \\ v_D & v_{D-1} & \dots & v_{D-L+1} \end{bmatrix} \begin{bmatrix} u_1 \\ u_2 \\ \vdots \\ u_L \end{bmatrix},$$

where $\textit{Toep}_0(\boldsymbol{u}) \in \mathbb{R}^{(D-L+1) \times D}$ and $\textit{Toep}(\boldsymbol{v}) \in \mathbb{R}^{(D-L+1) \times L}$. The validity of approximation (5) depends on the way $\widetilde{h}_m$ and $\widetilde{s}$ are sampled. In [26](Proposition 2), it is showed that if $\widetilde{s}(t)$ is band-limited with maximum frequency lower than $F_s/2$ and if we let the number of samples $N$ and the filter length $L$ grow to infinity, then model (5) is **exact** for the following sampling schemes:

$$\hat{s}(n) = \widetilde{s}(n/F_s), \ n \in \mathbb{Z} \tag{8}$$

$$\hat{h}_m(n) = (\widetilde{\phi} * \widetilde{h}_m)(n/F_s), \ n \in \mathbb{Z}. \tag{9}$$

Here, it is important to note that contrary to intuition, even in the idealized case where an infinite number of samples are available, the discrete-time filters $\{\hat{\boldsymbol{h}}_m\}_{m=1}^M$ involved in the measurement model are *never* streams of Diracs, but non-sparse, infinite-length filters consisting of decimated combinations of sinc functions. This is illustrated in Fig. 1. Recovering the original Dirac positions and coefficients from finitely many samples of such filters is a challenging task in itself.

### 2.1.2 Discrete frequency-domain model

Alternatively, one may approximate model (1) in the discrete finite-frequency domain. Let $\boldsymbol{x}_m \in \mathbb{C}^F$ denote the discrete Fourier transform (DFT) of $\hat{\boldsymbol{x}}_m$, defined by

$$x_m(f) = \text{DFT}(\hat{\boldsymbol{x}}_m) = \sum_{n=0}^{N-1} \hat{x}_m(n)e^{-2\pi i f n/F_s} \tag{10}$$

where $f$ belongs to a set of $F$ regularly-spaced frequencies $\mathcal{F} = \{f_1, \ldots, f_F\} \subset ]0, F_s/2]$ in Hz. We then have the following approximate model:

$$x_m(f) \approx h_m(f)s(f) \approx \left( \sum_{k=1}^{K} c_{m,k} e^{-2\pi i f \tau_{m,k}} \right) s(f) \tag{11}$$

where $\boldsymbol{h}_m \in \mathbb{C}^F$ and $\boldsymbol{s} \in \mathbb{C}^F$ denote the DFT of $\hat{\boldsymbol{h}}_m$ and $\hat{\boldsymbol{s}}$, respectively. Two approximations are made in (11). First, the time-domain convolution between $\hat{\boldsymbol{h}}_m$ and $\hat{\boldsymbol{s}}$ has been transformed into a multiplication through the DFT. This would be exact for a circular convolution, but the actual model is a linear convolution between infinite and non periodic signals, resulting in an approximation error. Second, the formula used for $\boldsymbol{h}_m$ in the right hand side of (11) is the one that would result from the discrete-time Fourier transform (DTFT) of $\hat{\boldsymbol{h}}_m$ which would require infinitely many samples $N$ to be calculated exactly, as opposed to the DFT. Note that the smoothing sinc kernel $\widetilde{\phi}(t)$ does not impact this formula, since only frequencies below $F_s/2$ are considered. Importantly, both approximations in (11) become arbitrarily precise as the number of samples $N$ grows to infinity.

While both the discrete-time model (5) and the discrete-frequency model (11) become increasingly accurate when $N$ becomes large, the latter directly incorporates the variables of interest $\{c_{m,k}, \tau_{m,k}\}_{m,k=1}^{M,K}$, as opposed to the former. In the remainder of this paper, it will be assumed that $\widetilde{s}(t)$ is bandlimited with maximum frequency less than $F_s/2$ and that $N$ is sufficiently large such that both models hold very well. This is a reasonable assumption in audio applications, where sensors typically acquire tens of thousands of samples per second. Moreover, we focus on situations where sensor noise is negligible. Hence, the approximation signs will be dropped for convenience.

## 2.2 Existing methods in channel identification

To the best of the authors' knowledge, all existing methods in blind channel identification rely on the discrete-time model (5) [12, 13, 14, 15, 16, 17, 18, 5, 19, 20]. The case of general emitted signals and finite filters was studied both methodologically and theoretically in the 90s [12, 13, 14], where two main categories of methods emerged, which we briefly review here, focusing on the two-channel ($M = 2$) case for simplicity. First, the so-called *subspace methods* rely on the estimation of a time-domain $MP \times MP$ covariance matrix where $P$ is a time-window length that must be larger than the filters' length $L$ [14]. The filters are estimated by spectral decomposition of this matrix. Second, the more common *cross-relation* (CR) methods rely on the observation that under noiseless conditions we have $\hat{\boldsymbol{h}}_m \star \hat{\boldsymbol{x}}_l - \hat{\boldsymbol{h}}_l \star \hat{\boldsymbol{x}}_m = \boldsymbol{0}_{N-L+1}$ for $l \neq m \in \{1, \ldots, M\}$, by associativity of the convolution. A common approach is therefore to solve a minimization problem of the form:

$$\hat{\boldsymbol{h}}_1^*, \hat{\boldsymbol{h}}_2^* = \underset{\hat{h}_1(1)=1}{\mathrm{argmin}} \left\| Toep(\hat{\boldsymbol{x}}_2)\hat{\boldsymbol{h}}_1 - Toep(\hat{\boldsymbol{x}}_1)\hat{\boldsymbol{h}}_2 \right\|_2^2, \tag{12}$$

which is a simple least-square problem. The constraint $\hat{h}_1(1) = 1$ is used to avoid the trivial solution $\hat{\boldsymbol{h}}_1 = \hat{\boldsymbol{h}}_2 = \boldsymbol{0}_L$. Alternatively, the normalization $\|\hat{\boldsymbol{h}}_1\|_2^2 + \|\hat{\boldsymbol{h}}_2\|_2^2 = 1$ can be used, leading to a minimum eigenvalue problem.

In the case of interest where the goal is to retrieve echo information from the filters, both subspace [17] and to a larger extent CR [15, 16, 18, 5] methods have been extended in order to handle sparse filters. This approach requires two independent steps: first estimating sparse filters, second retrieving echo locations and weights from them, typically using a peak-picking technique. Following the compressed sensing idea [21], sparsity is usually promoted using an $\ell_1$-norm penalty term on the filters. For instance in [16], the following LASSO-type [27] problem is considered:

$$\hat{\boldsymbol{h}}_1^*, \hat{\boldsymbol{h}}_2^* = \underset{\hat{h}_1(1)=1}{\mathrm{argmin}} \left\| Toep(\hat{\boldsymbol{x}}_2)\hat{\boldsymbol{h}}_1 - Toep(\hat{\boldsymbol{x}}_1)\hat{\boldsymbol{h}}_2 \right\|_2^2 + \lambda(\|\hat{\boldsymbol{h}}_1\|_1 + \|\hat{\boldsymbol{h}}_2\|_1) \tag{13}$$

and a Bayesian-learning method for the automatic inference of $\lambda$ is proposed. Several other approaches relying on similar schemes [15, 18, 5] have been proposed.

Four important bottlenecks of discrete-time methods for echo retrieval can be identified:

- Although they rely on sparsity-enforcing regularizers, the filters are strictly-speaking non-sparse in practice, due to the sinc kernel (Fig. 1). This general bottle-neck in compressed sensing has been referred to as *basis mismatch* and was notably studied in [28]. In particular, the true peaks of the filters do **not** correspond to the true echoes (Fig. 1), even for $N \to \infty$. Though, most existing methods rely on peak-picking [18, 5].

- For the same reason, these methods are fundamentally *on-grid*, namely, they can only output echo locations which are integer multiple of the sampling period $1/F_s$. This prevents subsample resolution, which may be important in applications such as room shape reconstruction from audio signals [3].

- These methods strongly rely on the knowledge of the length $L$ of the filters. However, due to the sinc kernel (Sec. 2.1.1), the true filters are always infinite.

- The dimension of the search space is $ML-1$, which is much larger in practice than the actual number $2MK$ of unknown variables. This makes the methods computationally demanding and sometimes intractable for large filter lengths (typically in the tens of thousands for acoustic applications).

## 3 Off-grid echo retrieval by multichannel annihilation

In this section, we introduce a novel method for echo recovery that makes use of the discrete-frequency model (11) and overcomes a number of shortcomings of existing approaches. Namely, it works directly in the parameter space, it does not rely on the filters' length but on the number of echoes, and it enables exact off-grid recovery of echoes' locations and weights in the noiseless case. The approach relies on the *finite rate of innovation* (FRI) sampling paradigm introduced in [23]. This is the first time this paradigm is applied to blind channel identification, to the best of the authors' knowledge.

### 3.1 The non-blind case

We start by considering the non-blind case where the emitted signal $s \in \mathbb{C}^F$ in the discrete frequency domain is known. We further assume throughout the paper that this signal is nonzero on the considered frequency grid $\mathcal{F} = \{f_1, \ldots, f_F\}$. We can then transform the discrete-frequency model (11) by writing:

$$h_m(f) = x_m(f)z(f) = \sum_{k=1}^{K} c_{m,k} e^{-2\pi i f \tau_{m,k}} \tag{14}$$

where the *Fourier-inverted* signal $z \in \mathbb{C}^F$ is defined by $z(f) = s(f)^{-1}$. Our goal is to estimate $\{c_{m,k}, \tau_{m,k}\}_{k=1}^{K}$ from $h_m = x_m \odot z_m$, where $\odot$ denotes the Hadamard product. If we take our frequency indexes $\mathcal{F}$ to be in arithmetic progression with step $\Delta_f$, then the exponential sequence $\{e^{-2\pi i f_i \tau_{m,k}}\}_{i=1}^{F}$ is a geometric progression with ratio $r_{m,k} = e^{-2\pi i \Delta_f \tau_{m,k}}$ for each $m, k$. Hence, $h_m$ is a weighted sum of geometric progressions. This enables us to use the so called *annihilating filter* technique [29]. This technique is based on the observation that

$$[1, -w] \star [w^0, w^1, w^2, \ldots, w^{F-1}] = \mathbf{0}_{F-1}, \tag{15}$$

for any $w \in \mathbb{C}$ and $F \in \mathbb{N}$. We deduce that if we define the filter $\boldsymbol{a}_m = [a_{m,0}, \ldots, a_{m,K}] \in \mathbb{C}^{K+1}$ as the following discrete convolution[1] of $K$ filters of size 2:

$$\boldsymbol{a}_m = [1, -r_{m,1}] \star [1, -r_{m,2}] \star \cdots \star [1, -r_{m,K-1}] \star [\mathbf{0}_{K-1}, 1, -r_{m,K}, \mathbf{0}_{K-1}], \tag{16}$$

then $\boldsymbol{a}_m$ is an *annihilating filter* for $h_m$, i.e., $\boldsymbol{a}_m \star h_m = \mathbf{0}_{F-K}$. Importantly, the number of echoes $K$ has to be known upfront in order to define $\boldsymbol{a}_m$. Let us now define the *polynomial representation* of filter $\boldsymbol{a}_m$ by:

$$P_{\boldsymbol{a}_m}[y] = \sum_{k=0}^{K} a_{m,k} y^k. \tag{17}$$

Because $\boldsymbol{a}_m$ is an annihilating filter for $h_m$, it follows from the classical interpretation of convolution as polynomial multiplication that $P_{\boldsymbol{a}_m}$ has exactly $K$ roots, which are the ratios $\{r_{m,k}\}_{k=1}^{K}$. Hence,

once an annihilating filter $\boldsymbol{a}_m$ for $\boldsymbol{h}_m$ has been found, the Dirac locations $\{\tau_{m,k}\}_{k=1}^K$ can be deduced by rooting $P_{\boldsymbol{a}_m}$. Once the roots are known, reconstructing the weights is a simple linear problem involving a Vandermonde matrix $\mathbf{V}(\boldsymbol{r}_m) \in \mathbb{C}^{F \times K}$, obtained by writing (14) in matrix form:

$$
\begin{bmatrix} h_m(f_1) \\ h_m(f_2) \\ \vdots \\ h_m(f_F) \end{bmatrix} = \begin{bmatrix} 1 & 1 & \cdots & 1 \\ r_{m,1}^1 & r_{m,2}^1 & \cdots & r_{m,K}^1 \\ \vdots & \vdots & \ddots & \vdots \\ r_{m,1}^{F-1} & r_{m,2}^{F-1} & \cdots & r_{m,K}^{F-1} \end{bmatrix} \mathbf{D}_m \begin{bmatrix} c_{m,1} \\ c_{m,2} \\ \vdots \\ c_{m,K} \end{bmatrix} = \mathbf{V}(\boldsymbol{r}_m)\mathbf{D}_m \boldsymbol{c}_m. \tag{18}
$$

where $\mathbf{D}_m = \mathrm{Diag}(e^{-2\pi i f_1 \boldsymbol{\tau}_m}) \in \mathbb{C}^{K \times K}$. The least-square solution of this system is given by $\boldsymbol{c}_m = \mathbf{D}_m^{-1}\mathbf{V}(\boldsymbol{r}_m)^{\dagger}\boldsymbol{h}_m$ where $\{\cdot\}^{\dagger}$ denotes the Moore-Penrose pseudo inverse. In practice, since positive weights are sought, the phases of this complex vector are discarded. General FRI theory [23] tells us that $F \geq 2K + 1$ is enough to uniquely recover the exact $K$ Dirac locations and weights using this method. In other words, the original echo retrieval problem has been reduced to that of finding an annihilating filter for $\boldsymbol{h}_m = \boldsymbol{x}_m \odot \boldsymbol{z}$. In practice, this can be done by solving the following minimization problem for $m = 1, \ldots, M$:

$$
\boldsymbol{a}_m^* = \underset{\|\boldsymbol{a}_m\|_2^2=1}{\mathrm{argmin}} \, \|Toep(\boldsymbol{x}_m \odot \boldsymbol{z})\boldsymbol{a}_m\|_2^2, \tag{19}
$$

where the unit norm constraint is used to avoid the trivial solution $\boldsymbol{a}_m = \boldsymbol{0}_{K+1}$. The solution of this problem is the eigenvector associated to the lowest eigenvalue of $Toep(\boldsymbol{x}_m \odot \boldsymbol{z})$. Assuming that the true $\boldsymbol{z}$ is given, that model (11) holds exactly and that $F \geq 2K + 1$, this eigenvalue will be unique and equal to 0.

## 3.2 MULAN: an iterative scheme

In the blind echo retrieval problem of interest, the emitted signal $\boldsymbol{s}$ and hence $\boldsymbol{z}$ are unknown. To solve for all unknown variables jointly, we introduce the following non-convex optimization problem:

$$
\boldsymbol{z}^*, \boldsymbol{a}_1^*, \ldots, \boldsymbol{a}_M^* = \underset{\|\boldsymbol{z}\|_2^2=\|\boldsymbol{a}_1\|_2^2=\cdots=\|\boldsymbol{a}_M\|_2^2=1}{\mathrm{argmin}} \sum_{m=1}^{M} \|Toep(\boldsymbol{x}_m \odot \boldsymbol{z})\boldsymbol{a}_m\|_2^2. \tag{20}
$$

Our strategy to tackle this problem is by alternated minimization with respect to each variable. Minimization with respect to each $\boldsymbol{a}_m$ is already covered by the previous section. Minimization with respect to $\boldsymbol{z}$ is also a minimum eigenvalue problem, since the cost function $C(\boldsymbol{z}, \boldsymbol{a})$ can be rewritten:

$$
C(\boldsymbol{z}, \boldsymbol{a}) = \sum_{m=1}^{M} \|Toep(\boldsymbol{x}_m \odot \boldsymbol{z})\boldsymbol{a}_m\|_2^2 = \sum_{m=1}^{M} \|Toep_0(\boldsymbol{a}_m)Diag(\boldsymbol{x}_m)\boldsymbol{z}\|_2^2 = \|\mathbf{Q}\boldsymbol{z}\|_2^2, \tag{21}
$$

where $\mathbf{Q} = [Toep_0(\boldsymbol{a}_1)Diag(\boldsymbol{x}_1); \ldots; Toep_0(\boldsymbol{a}_M)Diag(\boldsymbol{x}_M)] \in \mathbb{C}^{M(K+1) \times F}$ \qquad (22)

and $[\cdot; \cdot]$ denotes vertical concatenation. If the algorithm succeeds in bringing down the cost function to zero, it means that appropriate annihilating filters have been found for all channels for a given Fourier-inverted signal $\boldsymbol{z}$, and the locations and weights of all Diracs can be recovered. We call this method MULAN for *MULtichannel ANnihilation*. Pseudo-code for the algorithm is given in Alg. 1. Since (20) is non-convex, the alternate minimization scheme is at best guaranteed to converge to a stationary point of the cost-function $C(\boldsymbol{z}, \boldsymbol{a})$. To alleviate this issue, we propose to initialize the method multiple times with random values of $\boldsymbol{z}$ and only keep the run with lowest final $C(\boldsymbol{z}, \boldsymbol{a})$.

---

**Algorithm 1** MULAN (MULtichannel ANnihilation)

---

**Input:** Frequency-domain multichannel measurements $\{\boldsymbol{x}_{1:M}(f); f \in \mathcal{F}\}$ computed via DFT (10); *max_iter*; *conv_thresh*.

**Output:** Echo locations and weights $\{\tau_{m,k}, c_{m,k}\}_{m,k=1}^{M,K}$.

---

1: $iter := 0$; $\boldsymbol{z} := \text{random}()$;    *// i.i.d. standard complex Gaussian in $\mathbb{C}^F$*
2: **repeat**
3:    $iter := iter + 1$;
4:    **for** $m = 1 \to M$ **do:** $\boldsymbol{a}_m := \text{min\_eig\_vec}(Toep(\boldsymbol{x}_m \odot \boldsymbol{z}))$; **end for**
5:    $\boldsymbol{z} := \text{min\_eig\_vec}(\mathbf{Q})$;    *// See eq. (22)*
6: **until** $iter=max\_iter$ **or** $C(\boldsymbol{z},\boldsymbol{a})$ decreased by less than *conv_thresh*    *// See eq. (21)*
7: **for** $m = 1 \to M$ **do**
8:    $\boldsymbol{r}_m := \text{roots}(P\boldsymbol{a}_m)$; $\boldsymbol{\tau}_m := -\arg(\boldsymbol{r}_m)/(2\pi\Delta_f)$; $\boldsymbol{c}_m := \text{abs}(\mathbf{D}_m^{-1}\mathbf{V}(\boldsymbol{r}_m)^{\dagger}\boldsymbol{h}_m)$;  *// Sec. 3.1*
9: **end for**
10: **return** Shifted and scaled $\{\tau_{m,k}, c_{m,k}\}_{m,k=1}^{M,K}$;    *// See Sec. 3.3*

---

### 3.3 Identifiability and ambiguities

The identifiability of blind channel identification for general discrete filters and signals has been studied some time ago [12]. It is known that the filters $\{\hat{\boldsymbol{h}}_m\}_{m=1}^M$ cannot be recovered if their polynomial representations admit at least a common root or if the polynomial representation of the emitted signal $\hat{s}$ has less than $2L + 1$ roots. The latter is ruled out if the emitted signal has a rich enough spectral content (enough nonzero frequencies) which is usually the case for natural signals. The former has at least one consequence in our case: the problem is unidentifiable if the observed signals are scaled and delayed versions of each other, which may happen in practice. While other common roots may appear in the general setting, it is important to note that MULAN restricts the search of filters to those which are linear combinations of geometrical series in the frequency domain. There is no complete theoretical study on common roots in this case, to the best of the authors' knowledge. The authors of [30] theoretically studied blind deconvolution of sparse signals, but their results do not apply here since our filters are not sparse (see Sec. 2.2). Another well-known ambiguity is that the filters can only be recovered up to a global time-shift and scaling, because a converse shifting and scaling of the emitted signal yields the same observations. We handle this by adopting the convention $\tau_{1,1} = 0$ and $c_{1,1} = 1$. Additionally, we assume that all echoes are located in the first half of temporal filters to avoid time-wrapping ambiguities. Finally, the proposed MULAN algorithm has an extra specific ambiguity. It can be easily shown that multiplying the roots of all polynomials $\{P\boldsymbol{a}_m\}_{m=1}^M$ by a complex scalar $\gamma$ while dividing the Fourier-inverted signal $\boldsymbol{z}$ element-wise by a geometric series of ratio $\gamma$ does not change the cost function $C(\boldsymbol{z}, \boldsymbol{a})$. This can be handled by rescaling the roots of all annihilating filters to have unit modulus at each iteration. However, since only the complex arguments of the roots are used in the end, this appeared to be unnecessary in our experiments.

## 4 Experiments

### 4.1 On-grid vs. off-grid echo retrieval

We first emphasize the specific ability of the proposed method to recover echo locations off-grid by comparing it to conventional on-grid methods on a simulated room-acoustic scenario and on an artificial scenario with truly sparse discrete filters for reference. For the room-acoustic scenario, there is a point source emitting speech from the TIMIT dataset [31], and $M = 2$ microphones are randomly placed inside 100 random shoe-box rooms whose sizes vary from 4m × 6m × 8m to 5m × 7m × 9m. Simulations were performed using the *pyroomacoustics* library [32]. The absorption coefficient of each surface of the room is set to 0.2. Only first-order reflections on the 6 surfaces and the direct path are simulated, resulting in $K = 7$ echoes per channel and filters shorter than 50 ms. For each experiment, it was ensured that the minimum separation of echoes was 1ms. The filters are simulated in the continuous-time domain using the image-source method [33]. They are then smoothed, sampled and convolved with the source signal at $F_s = 16$kHz according to the measurement model described in Sec. 2.1. The ground-truth echo locations and weights are saved in the time-domain before smoothing and are hence off-grid. The $M$-channel input signals used are

0.25s long, *i.e.*, $N = 0.25F_s = 4000$ samples. On the other hand, for the artificial scenario, the speech source was discretely convolved with sparse filters of similar length with $K = 7$ nonzero elements each resulting in $N = 4000$ samples of $M$-channel observations. The ground-truth echo locations and weights are hence on-grid in this case. All weights take values between 0 and 1.

For MULAN, the DFT (eq. 10) is applied to each input signal using a grid $\mathcal{F}$ of $F = 401$ regularly spaced frequencies between 200 Hz and 2000 Hz. Such a choice of the frequency range avoids low-frequency bands which are often noisy in real scenario, while focusing on a typical spectral range for speech, but it can be easily adapted depending on the application. An odd number of frequencies was chosen, since it has proven to be a good practice [24]. We use 20 random initializations as a good compromise between global convergence and computational complexity, *max_iter*= 1000 and *conv_thresh*= 0.1%. The two baseline methods chosen are CR [12] as described in (12) and its LASSO-type extension [16] as described in (13). The filters lengths L were always set to the true lengths (which never exceed $0.05F_s$) and the sparsity parameter $\lambda$ for LASSO was manually set to $\lambda = 10^{-3}$, which empirically showed best performance among the choices $\{10^{-6}, 10^{-5}, \ldots, 10^2\}$, although any value below $10^{-2}$ showed similar performance.

We used two distinct metrics to evaluate Dirac location estimation and Dirac weight estimation. For the first one, a test is counted as successful if the root mean squared error (RMSE) of the $7 \times 2 = 14$ Dirac locations is below 1 sample ($1/F_s$ seconds), and the success rate out of 100 tests is provided. This metric only counts fully successful channel recovery and penalizes tests where some Diracs are missed or completely off. For the second one, we provide the weight RMSE of successful tests only. This is to avoid counting weights estimated at wrong Dirac locations. These metrics for 100 on- and off-grid tests and all three methods are showed in Table 1. We can see that for the on-grid case, both CR and MULAN perform well, CR even achieving more location recoveries than MULAN. This is not too surprising since CR is based on the on-grid artificial model, while MULAN uses an off-grid model. We observed that LASSO struggles with the proximity of Diracs and did not perform as well. In terms of weight estimation MULAN yields errors which are 2 to 3 orders of magnitudes smaller than the two competing methods, which is very encouraging. In the more realistic off-grid scenario, we observed that localization errors of CR and LASSO drastically degrades with almost no successful channel estimation. Meanwhile, MULAN achieves near-exact full recovery of locations and weights in 70 out of 100 tests.

## 4.2 Influence of $K$, $M$, $F$ on recovery rate

We now conduct further experiments to check the influence of parameters $K$, $M$ and $F$ on the ability of MULAN to fully recover Dirac locations and weights off-grid. We show results with 20 random initializations, $F = 201$ or $F = 401$ in the same frequency range as before, $M \in \{2, \ldots, 7\}$ and $K \in \{2, \ldots, 7\}$. The following RMSE thresholds were defined for success of recovery: 1 sample for locations as before and $10^{-2}$ for weights. 100 experiments were performed for every parameter set. Results for $F = 201$ can be seen in Figures 2 and 3, and for $F = 401$ in Figures 4 and 5. As can be seen, a higher recovery rate is generally observed when fewer echoes are present and more frequencies are used. On the other hand, the number of sensors does not significantly affect recovery performance. This is expected since $\mathcal{O}(KM)$ parameters are estimated from $\mathcal{O}(MF)$ observations. Increasing the number of random initializations also showed to increase success by alleviating the non-convexity of the problem, at the cost of an increased computational requirement.

| case | method | full location recovery | weight RMSE |
|---|---|:---:|:---:|
| *on-grid* | CR [12] | **92** % | 0.0390 |
| | LASSO [16] | 13 % | 0.155 |
| | MULAN (proposed) | 59 % | **0.00016** |
| *off-grid* | CR [12] | 1% | 0.0442 |
| | LASSO [16] | 2% | 0.0346 |
| | MULAN (proposed) | **70** % | **0.00048** |

Table 1: Ratio of full Dirac location recovery (RMSE < 1 sample = $1/F_s$ seconds) and weight RMSE (successful cases only) for three methods over 100 on-grid and 100 off-grid tests. Weights take values between 0 and 1.

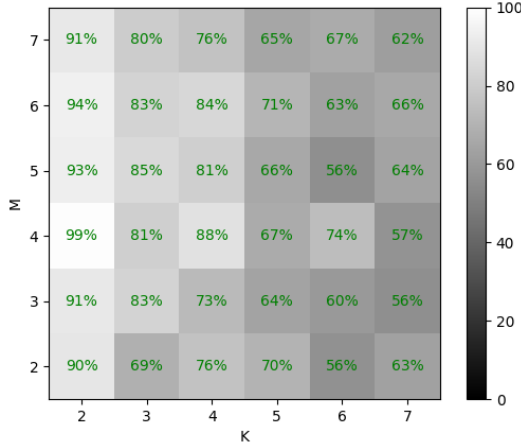
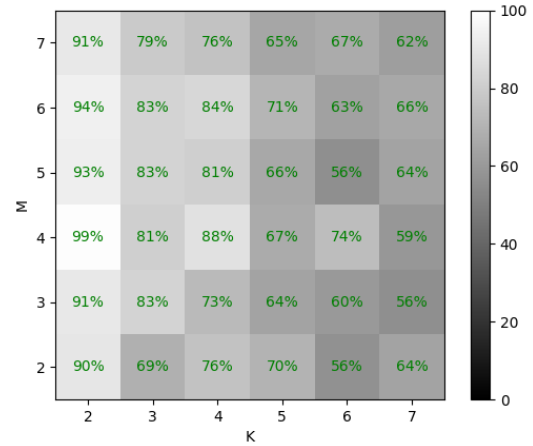

Figure 2: Rate of location retrieval for $F = 201$.

Figure 3: Rate of weight retrieval for $F = 201$.

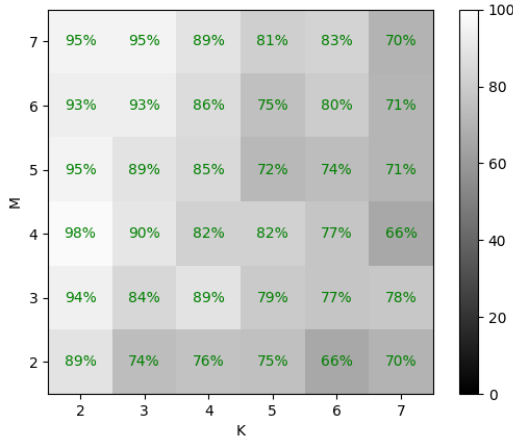
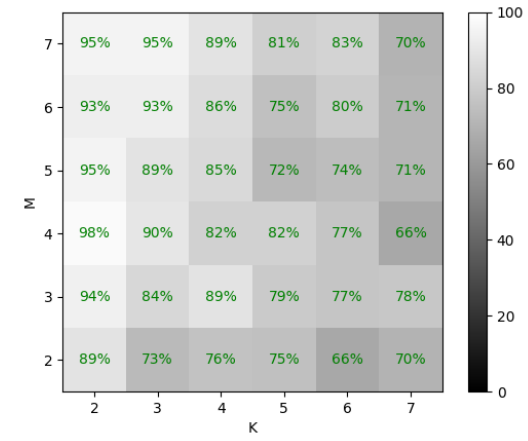

Figure 4: Rate of location retrieval for $F = 401$.

Figure 5: Rate of weight retrieval for $F = 401$.

## 5 Conclusion

This paper introduced the first method enabling blind and off-grid recovery of echo locations and weights from discrete-time multichannel measurements, to the best of the authors' knowledge. Future work will include alternative initialization schemes and convex relaxations in the spirit of [22] for the proposed cost function, extensions to sparse-spectrum signals and noisy measurements, and applications to dereverberation and audio-based room shape reconstruction. A better theoretical understanding of recovery guarantees as a function of $M$, $K$, $F$ and $N$ will also be sought. The code for this submission can be found at: `https://github.com/epfl-lts2/mulan`.

## Footnotes

[1] The chained discrete convolutions in (16) have to be taken from right to left to be compatible with (6).

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
