[Reviews · NeurIPS 2018]

Reviewer 1



This paper presents a method for estimating continuous-time delays for echos in synthetic multichannel impulse responses from underdetermined observations by doing so in the frequency domain. The fundamental trick enabling this technique to work is the "annihilating filter technique", by which a convolution in the frequency domain with a second-order polynomial is able to cancel out signals that are exponential in frequency, such as delays. This technique also applies for weighted sums of such signals, such as impulse responses made up of echoes. It is able to work on irregularly sampled frequencies (e.g., a subset of the frequencies computed by a discrete Fourier transform). Experiments are performed on synthetic impulse responses made up of pure delays with no additional noise and show that the proposed approach is significantly more accurate in identifying the locations and amplitudes of a known number of echoes than a LASSO-based approach and a standard cross-relation method. Other experiments show that the method is still able to identify the time of 46% of 7 impulses within 1us from 2-microphone recordings and that this ability improves with the number of microphones. Quality: The paper is technically sound. Its logic is relatively easy to follow, well presented, and seems to be correct. It discloses (but perhaps doesn't highlight) the assumptions that it makes in terms of the compositions of the impulse responses being analyzed (sums of pure impulses) and the necessity of providing the correct number of impulses in the impulse response. Experimental evaluation shows that it works very well within these constraints. Under-determined impulse response estimation is a hard problem and this approach is able to perform quite well for a constrained, but meaningful, subset of conditions. Clarity: The paper is very clearly written. It provides a thorough background to impulse responses and their estimation using clear notation and diagrams. The descriptions of "four important bottlenecks" of existing methods seems a little over-emphasized, in that the first two (non-sparse and on-grid) seem to be quite similar, and the third (strongly rely on knowledge of filter length) is similar to the proposed method's strong reliance on knowledge of the number of echoes. I believe that I would be able to reproduce the proposed approach from its description in this paper. Originality: The paper does a very good job of summarizing existing approaches to the problem. This approach of finite rate of innovation solutions to impulse estimation has recently been applied to other problems, but as far as I can tell has not been applied to impulse response estimation before. Significance: While the subset of noise-free recordings of a known number of perfectly specular and frequency-independent echoes is somewhat limited, it also provides a solid model of more realistic signals, thus paving the way for further development. It seems like this model should be robust to some amount of added noise, although it is not clear how robust it is to model mismatch or imperfect reflections. The proposed approach addresses a difficult task in a better way than previous work, thus advancing the state-of-the-art in a demonstrable way. The derived algorithm is a simple alternating minimization, where each step is a closed-form eigenvalue decomposition. The fact that it can be used on an arbitrary subset of frequencies from a DFT is also interesting and facilitates follow-up work. Minor comments: * Line 112: "the later" should be "the latter" * Equation (22): should Q be equal to the sum of the Toep_0(a_m)Diag(x_m) terms instead of their concatenation? I have read the authors' response.

Reviewer 2



I really liked this paper. The authors present an algorithm that perform blind echo estimation bypassing the messiness of discretization. The algorithm is explained very clearly, the problems are very well illustrated, and the approach is relatively easy to follow. Some nitpicks: – I would have appreciated a bit more discussion on the potential practical constraints that this method imposes. E.g., there is a mention that a long N is required, but it isn't clear what the practical limits of this are in a real audio problem. – As much as I like pyroomacoustics, it isn't the real thing. There is still the open question of whether this can work in real-life. There is certainly no shortage of papers that look good on paper, but never made it to the real-world. It helps to address this point with some experiments. The major issue: If this was a paper submitted to TSP or some other selective signal processing venue, I would enthusiastically support it; it is a very good paper. I am having some trouble though seeing how this submission fits at NIPS. There is arguably no machine learning or neuroscience in this paper. The only tenuous connection is the compressive sensing angle, but the narrow application domain is not something that would be if interest to the general practitioner (unlike most of the good CS papers I've seen at NIPS). Ultimately, I think this is a decision that should be left to the area editor. I think this is a good paper, but I do not think it belongs in NIPS. It would find a more sympathetic audience, and be more appreciated, in a signal processing venue.

Reviewer 3



Multi-channel blind deconvolution is a well-studied problem in the literature especially for the case of discrete-time signals and channels. The main novelty of this work is to consider a dispersive channels with taps with unknown delays and weight, where the traditional quantization (in the delay domain) results in a basis mismatch and degrades the performance. The authors have combined nonconvex methods, which exists for blind deconvolution in previous works such as "Fast and guaranteed blind multichannel deconvolution under a bilinear system model" by Kiryung Lee, Ning Tian, and Justin Romberg which also provides some initialization technique for the nonconvex optimization, by super-resolution methods (vis root finding). I have the following comments: 1. paper has a very marginal novelty, basically combining two already known methods, and the performance is shown to be good only empirically without any theoretical analysis. 2. there are some typoes here and there in the paper (FRI)sampling --> (FRI) sampling guaranteed to converged --> guaranteed to converge 3. the phase transtion plot in Figure 3 is a little bit confusing: increasing the number of delay taps degrades the performance as expected but increasing the number of channels "M" does not show any monotonicity? 4. It seems that the authors have assumed that the number of delay taps $K$ is exactly known for the recovery algorithm. What happens when $K$ is unknown? In super-resolution techniques such as TV-norm minimization this is not a big issue since the number of taps can be also extracted. Is it also true for the proposed algorithm? It would be good if the authors consider the general case for doing the simulations and report the results.